# MiR-126-Loaded Immunoliposomes against Vascular Endothelial Inflammation In Vitro and Vivo Evaluation

**DOI:** 10.3390/pharmaceutics15051379

**Published:** 2023-04-30

**Authors:** Yongyu Tang, Ying Chen, Qianqian Guo, Lidan Zhang, Huanhuan Liu, Sibu Wang, Xingjie Wu, Xiangchun Shen, Ling Tao

**Affiliations:** 1The State Key Laboratory of Functions and Applications of Medicinal Plants, School of Pharmaceutical Sciences, Guizhou Medical University, University Town, Guian New District, Guiyang 550025, China; 2The Department of Pharmacology of Materia Medica (The High Efficacy Application of Natural Medicinal Resources Engineering Center of Guizhou Province and The High Educational Key Laboratory of Guizhou Province for Natural Medicinal Pharmacology and Druggability), School of Pharmaceutical Sciences, Guizhou Medical University, Guiyang 550031, China; 3The Key Laboratory of Optimal Utilization of Natural Medicine Resources (The Union Key Laboratory of Guiyang City-Guizhou Medical University), School of Pharmaceutical Sciences, Guizhou Medical University, Guiyang 550031, China; 4The Key Laboratory of Endemic and Ethnic Diseases of Ministry of Education, Guizhou Medical University, Guiyang 550004, China

**Keywords:** vascular endothelial inflammation treatment, miR-126 delivery, VCAM-1 targeting, cationic liposomes

## Abstract

Due to the accompaniment of vascular endothelial inflammation during the occurrence and development of cardiovascular diseases (CVD), treatment modalities against vascular endothelial inflammation have been intensively investigated for CVD prevention and/or treatment. Vascular cell adhesion molecule-1 (VCAM-1) is a typical transmembrane inflammatory protein specifically expressed by inflammatory vascular endothelial. By inhibiting VCAM-1 expression through the miR-126 mediated pathway, vascular endothelial inflammation can be efficiently relieved. Inspired by this, we developed a miR-126-loaded immunoliposome with VCAM-1 monoclonal antibody (VCAM_ab_) decorated at its surface. This immunoliposome can be directly targeted to VCAM-1 at the inflammatory vascular endothelial membrane surface and achieve highly efficient treatment against inflammation response. The cellular experiment results showed the immunoliposome had a higher uptake rate towards inflammatory human vein endothelial cells (HUVECs) and can significantly downregulate the VCAM-1 expression level of inflammatory HUVECs. In vivo investigation further demonstrated that this immunoliposome displayed a higher accumulation rate at vascular inflammatory dysfunction sites than its non-VCAM_ab_-modified counterpart. These results suggest that this novel nanoplatform can effectively deliver miR-126 to vascular inflammatory endothelium, opening a new avenue for the safe and effective delivery of miRNA for potential clinical application.

## 1. Introduction

Despite great advancements in cardiovascular disease (CVD) treatment made in recent years, mortality caused by CVD has accounted for more than 30% of global deaths according to the statistics of the World Health Organization [1]. Currently, most anti-CVD drugs in clinics suffer from long treatment periods, systemic side effects and low patient compliance. Therefore, highly specific CVD treatment modality with low side effects has drawn great attention. Vascular endothelial cell refers to a single-layer flat epithelium lining the inner surface of blood vessels and lymphatic vessels. Pathological conditions such as blood flow scouring, high blood glucose level and high blood lipid level would damage vascular endothelial cell structure and cause serious vascular inflammation response, which would deteriorate into various cardiovascular diseases including hypertension, atherosclerosis and coronary heart disease [2,3]. Further investigations proved that the CVD development would in turn intensify vascular endothelial inflammation. Therefore, treating vascular endothelial inflammation is an efficient and specific measure to prevent and treat malignant CVD [4,5,6].

Vascular cell adhesion molecule-1 (VCAM-1) is a cell surface sialoglycoprotein belonging to the immunoglobulin family. As a pro-inflammatory cytokine, VCAM-1 is specifically expressed by endothelial cells several hours after exposure to cytokines such as TNF-α and IL-1β. VCAM-1 can activate and promote various critical pathological processes in the development of CVD, including monocyte adhesion and platelet aggregation at the vascular injury site [7,8]. The concentration of VCAM-1 is positively correlated with CVD severity, and the downregulation of VCAM-1 can dramatically relieve vascular endothelial inflammation responses. These unique properties make VCAM-1 an ideal target for the treatment of inflammation.

MiRNA is a type of non-coding small RNA with 17~25 bases capable of regulating protein expression by interfering with gene transcription. According to the literature, there are about 1000 kinds of miRNA sequences, which can regulate about 30% of human gene expression involved in a series of important physiological and pathological processes [9,10]. Recently, miRNA has been widely applied in the development of technologies for CVD diagnosis, prevention and treatment [11,12]. MicroRNA-126 (miR-126) is an endothelial-specific RNA sequence and can suppress VCAM-1 expression with the help of interferon regulatory factor-1 [13]. Moreover, miR-126 can mitigate CVD by adjusting MARK and P13K-mediated angiogenesis [14,15,16]. In consequence, miR-126 can be applied to construct treatment modalities against vascular endothelial inflammation [17,18]. However, the current miRNA delivery system faces obstacles including poor cell uptake rate, enzyme-mediated RNA degradation and limited accumulation at the lesion site, which significantly limits its clinical translation [19]. To achieve high therapeutic efficiency of RNAs, precise delivery and accumulation of RNAs at targeted tissue/organ are of importance for delivery systems. Therefore, it is urgent to develop a delivery nanoplatform capable of enhancing RNA uptake for disease treatment. 

Compared to various viral vectors used for gene delivery, cationic liposomes displayed enhanced targeting properties with higher biocompatibility. Additionally, cationic liposomes can be produced on a large scale with excellent reproducibility [20]. However, cationic lipids are highly cytotoxic [21]. In order to reduce cationic lipids cytotoxicity and improve the transfection efficiency of siRNA, biocompatible materials, such as polyethylene glycol, hyaluronic acid and hydrophilic peptides, were frequently attached to cationic liposomes surface [22]. Cationic liposomes can not only effectively protect genes from enzymatic degradation but also enable receptor-ligand interaction-mediated drug accumulation towards target tissues/organs. As a result, promoted cellular internalization and gene transfection can be achieved specifically to target tissues with enhanced expression of specific receptors [23,24]. Therefore, cationic liposomes are ideal nanocarriers for gene delivery.

Due to its low toxicity, non-immune responsiveness and high specificity, the antibody has been intensively applied as a targeting ligand for constructing active targeting drug delivery nanocarriers, such as antibody-decorated liposomes [25,26,27,28] and antibody–drug conjugates (ADC) [29,30,31]. ADCs have gained tremendous momentum in the last decade due to the FDA approval of four ADC applications [32,33,34,35]. However, existing gene-drug delivery systems are mainly used for the treatment and amelioration of tumors [36,37,38,39] and rarely for the treatment of cardiovascular diseases.

In this article, we prepared miR-126-loaded cationic liposomes through a thin film dispersion extrusion method, which was further modified with VCAM_ab_ on the surface through an amidation reaction. After being intravenously injected, the lipid layer of the cationic liposomes can physically preserve miR-126 from enzymatic degradation for enhanced bioavailability. Moreover, the liposome can accumulate more at the thoracic aortic inflammation site inducing LPS under the guidance of VCAM_ab_. Overall, we developed a highly specific and efficient miRNA delivery nanoplatform capable of mitigating inflammation response and potentially treating CVD of various kinds. 

## 2. Materials and Methods

### 2.1. Materials

The (2,3-Dioleoyloxy-propyl)-trimethylammonium-chloride (DOTAP) was purchased from MedChem Express (Shanghai, China). 1,2-Distearoyl-sn-glycero-3-phosphoethanolamine-N-[methoxy(polyethylene glycol)-2000] (DSPE-mPEG2000) was purchased from Shanghai Shunna Biotechnology Co., Ltd. (Shanghai, China). DSPE-Poly(ethylene glycol)-succinimides ester, (DSPE-PEG2000-NHS) average MW2000 was purchased from Nano soft Biota Chaology LLC (Shanghai, China). Cholesterol and anhydrous ethanol were purchased from Sino Pharm Chemical Reagent Co. Ltd. (Shanghai, China). Diethyl pyro carbonate (DEPC) and Green fluorescent nucleic acid dye were purchased from Beijing Solarbio Biotechnology Co., Ltd. (Beijing, China). MicroRNA-126 mimics, cy3-miR-126 and cy5-miR-126 were purchased from Shanghai GenePharma Co., Ltd. (Shanghai, China). Agarose was purchased from Beijing Yami Biotechnology Co., Ltd. (Beijing, China). VCAM_ab_ and β-actin antibody (β-actin_ab_) were purchased from Affinity Biosciences. Thiazolyl blue tetrazolium bromide (MTT) was obtained from Aladdin (Shanghai, China). The 4’,6-Diamidino-2-phenylindole (DAPI) was purchased from Keygen Biotech (Nanjing, China). Elisa Kits (TNF-α, IL-1β and IL-6) were purchased from Nanjing Jiancheng Bioengineering Institute. (Nanjing, China). RPMI-1640 Medium, fetal bovine serum, penicillin, streptomycin and pancreatic enzyme were purchased from Gibco (Grand Island, NV, USA). The BCA protein assay kit was purchased from Beijing Solarbio Biotechnology Co., Ltd. (Beijing, China).

### 2.2. VCAM-1 Expression in LPS-Activated HUVECs Cells 

The expression of transmembrane VCAM-1 protein was determined using Western blot analysis [40]. HUVECs were cultured in a 6 cm Petri dish (5 × 10^6^/well) until 80% confluence was achieved. Then, HUVECs were washed with PBS and treated with LPS with 0.1–10 μg/mL final concentration at 37 °C for 8 h, followed by washing with PBS and lysing in 100 μL RIPA lysis buffer (50 mM Tris pH 7.4, 150 mM NaCl, 1% Triton X-100, 1% sodium deoxycholate, 0.5% SDS) (Beyotime Biotechnology, Shanghai, China) containing 1% PMSF for 25 min on ice. The cell lysate was centrifuged with a speed of 12,000× *g* for 20 min at 4 °C. The protein concentration of the supernatant was detected by the BCA Protein Assay Kit and diluted to the same concentration prior to Western blotting analysis. These collected supernatants were resolved on 10% SDS-PAGE (Beijing, China) and transferred to PVDF membranes (Microtiter wells) for blotting. The membranes were closed with TBST buffer containing 5% BSA and then incubated overnight at 4 °C with 1:1000 dilution of VCAM_ab_ and 1:5000 dilution of β-actin_ab_ in Tris-buffered saline containing 0.1% Tween 20. Horseradish peroxidase-conjugated secondary antibodies were then used. The membranes were washed and stained for gel imaging analysis (Bio-rad, USA) using β-actin as the internal reference. Densitometry analysis was performed by Imaging software (Version 5. 2 build 14).

### 2.3. Preparation of CLs and Va-CLs

The cationic liposomes (CL) were prepared by a thin film dispersion extrusion method [41]. Briefly, lipids consisting of DOTAP, Chol, and 0.5% mPEG2000 DSPE or NHS-PEG2000-DSPE were dissolved in chloroform with a molar ratio of DOTAP to Chol of 3/1. Thin lipids were formed by removing chloroform in a rotary evaporator at 45 °C and then drying under vacuum for 30 min. The membranes were then hydrated with 1.5 mL PBS buffer (pH 7.2–7.4) or 1.5 mL PBS buffer (pH 7.2–7.4) containing VCAM_ab_ (9.09 μg/mL) for 1 h at 50 °C. The reaction was carried out overnight at 4 °C. Subsequently, the liposome protoplasts were extruded through 100 nm pore-sized polycarbonate nanopore membranes to obtain CLs and Va-CLs.

### 2.4. VCAM_ab_ Coupling Efficiency (VCAM_ab_%) 

VCAM_ab_ conjugation efficiency (VCAM_ab_%) to CLs was determined using the BCA protein kit [42]. Briefly, the working reagent was prepared through the mixture of reagents A and B, at the ratio of 50:1. A total of 96-well microplates, 20 µL of each albumin standard (0.05–0.5 µg/mL) or 20 µL of Va-CLs before and after dialysis, were added. To each well, 200 µL of working reagent was added and the microplate was homogenized for 30 s and incubated at 37 °C for 30 min. The absorbance was read at 570 nm in a microplate reader. The conjugation efficiency was calculated as the percentage ratio of antibody quantified in Va-CLs to the total antibody employed initially for conjugation.

### 2.5. VCAM_ab_ Conjugate Evaluation by SDS-PAGE Electrophoresis

Antibody integrity on the surface of Va-CLs and CLs was studied by polyacrylamide gel electrophoresis (SDS-PAGE) under denaturing conditions [42]. The samples were first incubated with protein loading buffer for 5 min at 95 °C. Then, a Western blot electrophoresis chamber (Bio-Rad, Hercules, CA, USA), 10% SDS-PAGE and Tris/glycine/SDS running buffer were used. Electrophoresis was performed for 90 min at 120 V, using protein markers as protein ladders. After staining with Coomassie brilliant blue, the gels were washed with H_2_O and then photographed.

### 2.6. Agarose gel Electrophoresis

Agarose gel electrophoresis was performed to study the loading of miR-126 by Va-CLs and CLs. The miR-126 and Va-CLs or CLs were mixed well at DOTAP:miR-126 molar ratios of 16:1, 24:1, 32:1, 40:1, 48:1 and 56:1, and then incubated for 30 min at room temperature to obtain Va-CLs/miR-126 and CLs/miR-126 complexes. The above two samples were electrophoresed on a 1% (*w*/*v*) agarose gel containing green fluorescent nucleic acid dye. Electrophoresis was performed for 30 min at 120 V using 1× TAE buffer and imaged using a gel imaging system (Bio-Rad, USA).

### 2.7. RNase Protection Assay

RNase protection assay was performed according to Perche et al. Samples containing miR-126 were incubated with 0.25 mg/mL RNase A (Beijing Solarbio Biotechnology Co., Ltd., Beijing, China) for 3 h at 37 °C. The RNase was then inactivated with 70% ethanol before complex dissociation using 1% Triton 100. Then, samples were analyzed on a 1% agarose gel containing ethidium bromide. Gel was imaged using a gel imaging system (Bio-Rad, USA).

### 2.8. Va-CLs/miR-126 and CLs/miR-126 Characterization

#### 2.8.1. Transmission Electron Microscopy

A total of 5 μL of the diluted Va-CLs/miR-126 and CLs/miR-126 was dropped onto the copper grid and allowed to stand in place for 5 min. The excess was drained using filter paper, and the copper grip was dried. Then, 5 μL of 1% phosphotungstic acid was added. Staining lasted 5 min, and the copper grid was dried with filter paper at room temperature. Images of the nanostructures in the solution states were captured using a transmission electron microscope (TEM, JOEL 2100f, Tokyo, Japan) at an accelerating voltage of 120 kV. 

#### 2.8.2. Size and Zeta Potential

The size of four samples was determined using dynamic light scattering (DLS) on a Nano Brook 90 Plus PALS (Brookhaven Instruments, Holtsville, NY, USA) equipped with a 640 nm laser. Measurements were performed under automatic attenuator selection at a 10 µM total lipid concentration in water at 25 °C with the following physico-chemical parameters: viscosity of dispersant (water) 0.890 cP, dispersant dielectric constant 78.54, refractive indices: 1.525 for Chol and 1.331 for dispersant. One individual record represents the average of 3 measurements per sample and for each sample, three such records were acquired. The reported Z-averages of intensity distributions (in nm) represent averages of these measurements, together with the PDI (polydispersity index), which reflects the homogeneity of the sample.

For Zeta potential measurements, a Universal Dip Cell (SR-0009) was immersed into the sample after size measurement. The Zeta potential was determined using electrophoretic light scattering (ELS) under automatic voltage selection with 60 s delay between measurements via the Smoluchowski model. Individual records represent the average of 5 measurements and for each sample, three such records were acquired. The results (expressed in mV) were presented as the averages of these measurements, together with the standard deviation. The results were analyzed using the built-in Nano Brook 90Plus PALS (Brookhaven Instruments).

### 2.9. Cell Culture

The human umbilical vein endothelial cells (HUVECs) were purchased from American Type Culture Collection (ATCC). The cells were cultured in Roswell Park Memorial Institute-1640 Medium (RPMI-1640) containing 10% FBS and 100 U/L penicillin-streptomycin in a 37 °C cell incubator with 5% CO_2_.

### 2.10. Confocal Microscope

HUVECs were plated on 6-well chamber slides at a density of 2 × 10^5^ cells per well and cultured for 24 h at 37 °C. After discarding the upper medium and washing twice with PBS, 1 μg/mL LPS was added to HUVECs cells, followed by incubation for 8 h at 37 °C. Then, the cells were incubated with cy5-miR-126-loaded liposomes with equivalent cy5-miR-126 concentration (50 nM and 75 nM). After being incubated together for 3 h, the cells were washed with PBS and fixed with 4% paraformaldehyde for 20 min. After further incubating the fixed cells with DAPI (5 μg/mL) for nucleic acid staining, the cover slides were placed on a sticky microscope slide in the presence of an anti-fluorescence quenching agent. The cy5-miR-126 fluorescence intensities were observed and analyzed using an Olympus FV-3000 laser scanning microscope equipped with FLUOVIEW software (Version 2.5.1.228) (Olympus, Tokyo, Japan) 

### 2.11. Flow Cytometry

For the flow cytometry analysis, HUVECs (1 × 10^6^) were cultivated in 12-well plates for 24 h, followed by incubation with LPS for 8 h. Then, the cells were treated with cy.3-miR-126- (50 nM and 75 nM) loaded liposome for 3 h. The receptor inhibition assay was treated with 1 μg/mL LPS for 8 h. The cells were first co-incubated with basal medium containing 0.01 mg/mL VCAM_ab_ for l h to fully saturate the cell surface with VCAM-1 receptors. Va-CLs/cy3-miR-126 and CLs/cy3-miR-126 (miR-126 concentration of 50 nM) were prepared separately and added to HUVECs for a 3 h incubation. The treated cells were then rinsed in trypsin solution and collected for the flow cytometry analysis (λ_ex_ = 550 nm, λ_em_ = 570 nm).

### 2.12. In Vitro Cytotoxicity Assay

HUVECs were seeded into 96-well plates (BIOFIL JET, Guangzhou, China) at a density of 2 × 10^5^ cells per well and cultured for 24 h. Afterward, the cells were incubated with Va-CLs/miR-126, CLs/miR-126 or free miR-126 with equivalent miR-126 concentration. After incubation for 8 h, the cell viability was measured using an MTT assay. 

### 2.13. Western Blotting

After seeding 5 × 10^6^ HUVECs into 6 cm Petri dishes, Va-CLs/miR-126, CLs/miR-126, free miR-126, Va-CLs or CLs with equivalent miR-126 concentration and 1 μg/mL LPS were added to the culture medium and incubated with HUVECs at 37 °C for 8 h. Then, the same steps as described in Section 2.2 were applied for Western blotting analyses. 

### 2.14. Detection of VCAM-1 mRNA by qPCR

Total RNA was extracted from HUVECs by the RNA prep Pure Total RNA Extraction Kit for Cultured Cells/Bacteria (Yaanta Biotechnology Co., Ltd., Beijing, China) according to the manufacturer’s protocol. An equal amount of RNA (2 μg) was reverse transcribed to cDNA using the Hifair III 1st Strand cDNA Synthesis Kit (Yeasen, Shanghai, China). A quantitative real-time PCR (Heiff UNICON qPCR SYBR Green master Mix, Yeasen, Shanghai, China) was performed in a real-time PCR System (Bio-Rad, Waltham, MA, USA). The VCAM-1 mRNA-specific primers were used for the analysis of the expression of the VCAM-1 mRNA gene. The primers were synthesized by using Invitrogen Biotechnology Co., Ltd. (Shanghai, China) and the sequences were as follows: β-actin, forward 5′-CCTGGCACCCAGCACAAT-3′ and reverse 5′-GGGCCGGACTCGTCATAC-3′; VCAM-1, forward 5′-GGGAAGATGGTCGTGATCCTT-3′ and reverse 5′-TCTGGGGTGGTCTCGATTTTA-3′. The PCR cycle was performed as follows: stage 1, 1 cycle at 95 °C for 5 s; stage 2, 55 cycles at 95 °C for 5 s, and 60 °C for 30 s. At the end of the PCR reactions, the melt curve analyses were performed for all genes. The fold increase or decrease was determined relative to a blank control after normalizing to β-actin in each sample using the delta-delta Ct method.

### 2.15. Immunofluorescence

After the cells were treated according to the steps described in Section 2.13, 4% paraformaldehyde was fixed for 15 min, 0.2% Triton-100 was permeabilized for 10 min, and the cells were blocked for 30 min with the primary antibody added dropwise and incubated at 4 °C overnight. The next day, the primary antibody was recovered and washed twice with PBS, and the secondary antibody with fluorescence was added uniformly into each well under light-proof conditions and incubated for 1 h at room temperature.

### 2.16. Determination of Cytokine Secretion by ELISA 

HUVECs were seeded into 96-well plates at a density of 1 × 10^4^ cells/well. After the treatment with LPS was followed by Va-CLs/miR-126, CLs/miR-126, free miR-126, Va-CLs or CLs with equivalent miR-126 concentrations, the expression levels of IL-1β, IL-6 and TNF-α in the culture medium were detected using the ELISA kit by following the manufacturer’s instructions.

### 2.17. Animal Experiments

#### 2.17.1. Ethical Statement on Animal Experiments

Five-week-old male Kunming mice (18 ± 2 g) were purchased from the Experimental Animal Center of Guizhou Medical University. All the mice were raised under SPF conditions for a week before the experiments. All experiments and procedures were approved by the Animal Welfare and Ethics Committee of Guizhou Medical University (No: 2000664) and conducted according to the Regulations of Experimental Animal Administration issued by the State Committee of Science and Technology of China. 

#### 2.17.2. Analysis of Inflammation in Mice

After one week of adaptive feeding, 24 Kunming mice were randomly divided into 4 groups: normal control group (Control), model control group (LPS), Va-CLs/miR-126 group (Va-CLs/miR-126, 20 μg/each), CLs/miR-126 group (CLs/miR-126, 20 μg/each). The mice in the normal control and model control were injected with saline in the tail vein, and the mice in the remaining groups were injected with the corresponding drugs in the tail vein. The drugs were administered once every other day, and LPS (10 mg/kg) was injected intraperitoneally 12 h after the second dose, and the third dose was administered 12 h later. After 1 h, blood was taken from the orbit, the mice were executed by breaking their necks, and the thoracic aorta was removed for further analysis.

Mouse sera were tested promptly or stored at −20 °C for testing, and the expression levels of pro-inflammatory cytokines, including IL-1β and IL-6, were measured in serum according to the steps in the QuantiCyto^®^ ELISA assay kit (Neobioscience Technology Company, Beijing, China) instructions.

The thoracic aortic samples were randomly selected from three mice in each group for pathological observation. The thoracic aorta of each group was fixed in 4% paraformaldehyde for 24 h. The fixed tissue was then removed, after which the following steps took place: gradient dehydration, paraffin embedding, sectioning, dewaxing, rehydration, and finally HE staining. Next, immunohistochemistry (IHC) was employed to determine the expression level of VCAM-1 in the thoracic aorta of each group.

#### 2.17.3. The Inflammatory Injury Site of the Main Thoracic Vessels in Mice Targeting the Effects of Va-CLs/miR-126

To evaluate the VCAM_ab_ guided biodistribution of immunoliposome, Kunming mice were randomly divided into two groups and treated intraperitoneally with 10 mg/kg LPS for 12 h to induce verification injury. Then, the mice were intravenously injected with Va-CLs/miR-126 and CLs/miR-126 loading cy5 labeled miR-126. Three hours after injection, all the mice were sacrificed to collect major tissues (thoracic aorta, heart, liver, spleen, lung, kidney and brain) for ex vivo imaging using IVIS Spectrum CT (Perkin Elmer, Waltham, MA, USA).

### 2.18. Zebrafish In Vivo Imaging System Examines Targeting of Vascular Endothelial Inflammatory Injury Sites 

The model organism zebrafish (Wild Type, WT) wild-type zebrafish Tübingen strain, provided by the National and Local Joint Engineering Laboratory of Cell Engineering and Pharmaceutical Technology of Guizhou Medical University, Certificate of Conformity No. SYXK (Guizhou) 2018-0001. The fish eggs were placed in an incubator at a constant temperature, and the embryo water was changed every day and cultured with PTU embryo water after 6 h. After 3–5 days of incubation, zebrafish that were in good condition and similar in size were selected for the experiment.

Zebrafish were injected with 2 nL of LPS (0.5 mg/mL) in the yolk sac for 12 h. Zebrafish showed inflammatory responses such as spinal curvature, yolk wrinkling, cardiac hemorrhage, and pericardial enlargement. Va-CLs/cy5-miR-126 and CLs/cy5-miR-126 were injected into the LPS-inflamed zebrafish through the tail vein. Five fish in each group were anesthetized and placed in carboxymethyl cellulose after administration. The fish were collected at different time points (0.5, 1, 3, 6, 12, 24 and 48 h) following administration using a stereo fluorescence microscope.

### 2.19. Statistical Analysis 

The mean and standard deviations (S.D.) were determined with at least three independent experiments (n). Student’s *t*-test was used to compare the means of two groups, differences were considered statistically significant for values of *p* < 0.05.

## 3. Result

### 3.1. VCAM-1 Expression in LPS-Activated HUVECs Cells

As VCAM-1 played an indispensable role in targeted miR-126 delivery, we examined the VCAM-1 expression levels of HUVECs with various inflammation degrees (Figure 1). After HUVECs were treated with different concentrations of LPS for 8 h, the expression levels of VCAM-1 were significantly upregulated compared to the control group, which was in accordance with the published literature [43,44,45]. Therefore, 1 μg/mL LPS was selected as the modeling condition for subsequent experiments to ensure high expression of VCAM-1 in HUVECs cells.

### 3.2. Synthesis and Preparation of the Va-CLs Delivery System

To improve the efficiency of microRNA delivery specificity and produce more effective intracellular silencing activity of the corresponding genes, we decorated VCAM_ab_ at the surface of the cationic liposomes (Figure 2A). The functionalized phospholipid DSPE-PEG-NHS was added to the cationic liposomes to provide activated esters, which further reacted with the primary amino group to covalently anchor VCAM_ab_ at the liposome surface. The coupling efficiency was found to be 66.05%, measured from the purified fraction of Va-CLs. The coupling effect of VCAM_ab_ on CLs was confirmed using SDS-PAGE electrophoresis under reducing conditions. After reduction treatment, the spatial structure of VCAM_ab_ was destroyed because the disulfide bond in the hinge region of VCAM_ab_ was broken. However, the VCAM_ab_ primary structure was retained with the heavy chain (Hc) and light chain (Lc) being preserved. In addition, the band around 75 kD was a dimer that had not been denatured completely. Notably, the band between 63–75 kD represented BSA, which was added to the antibody by the manufacturer. The Va-CLs/miR-126 and Va-CLs showed corresponding bands at the heavy and light chains as free VCAM_ab_ (Figure 2B). However, due to the presence of pegylated long chains in the Va-CLs, the molecular weight of VCAM_ab_ increased after coupling with CLs. As a result, the VCAM_ab_ bands moved up slightly in comparison to the free VCAM_ab_. However, CLs/miR-126 and CLs without coupled antibodies did not show corresponding bands, indicating that VCAM_ab_ was successfully connected to the CLs.

### 3.3. Agarose Gel

To confirm the successful loading of miR-126 within liposome and optimize the Va-CLs formula for maximized miR-126 loading efficacy, the lipid complex was analyzed using gel retardation assay. In the process of electrophoresis, negatively charged miR-126 moved towards the positive pole, and the complex of miR-126 and cationic liposomes stayed in the loading pore. MiR-126 bands were observed for Va-CLs/miR-126 with a low DOTAP/miR-126 molar ratio, indicating insufficient miR-126 protecting effect (Figure 3A,B). In addition, gradually attenuated miR-126 bands were displayed with increasing DOTAP/miR-126 molar ratio, and no clear miR-126 band was observed for liposomes with DOTAP/miR-126 molar ratio higher than 32/1, which demonstrated that miR-126 was totally complexed by Va-CLs. The ability to protect miR-126 from enzymatic degradation is an important feature of an ideal nucleic acid therapeutic delivery system (Figure 3C). Neglect miR-126 signal was observed after RNase treatment for free miR-126, indicating that miR-126 was almost completely degraded. On the contrary, a strong miR-126 band appeared for Va-CLs/miR-126 complex, which indicates that miR-126 can be protected from enzymatic degradation after forming a complex with cationic liposomes. 

### 3.4. Va-CLs/miR-126 Preparation and Characterization 

The Tyndall effect of Va-CLs and CLs was observed under light beam irradiation (Figure 4A). The morphology of CLs/miR-126 and Va-CLs/miR-126 in a dry state was observed using a transmission electron microscope (Figure 4B,C). CLs/miR-126 and Va-CLs/miR-126 presented a regular spherical shape, and the surface of Va-CLs/miR-126 had irregular small protrusions. The hydration diameter and surface potential of liposomes were analyzed using a DLS instrument. The CLs had a hydration diameter of ~126 nm and a zeta potential of ~16 mV. After being decorated with VCAM_ab_, the CLs diameter increased to ~135 nm and a zeta potential of ~10 mV. When the molar ratio of DOTAP:miR-126 was 40:1, the size of CLs/miR-126 was ~160 nm, making it slightly bigger than the CLs. In addition, probably a lower zeta potential of CLs/miR-126 was confirmed after complexing with negatively charged miR-126, but it was not shown in the results. The size of Va-CLs/miR-126 was ~162 nm, making it slightly bigger than the Va-CLs. Additionally, a lower zeta potential of Va-CLs/miR-126 was confirmed after complexing with negatively charged miR-126 (Table 1). 

### 3.5. In Vitro Targeting Study of Va-CLs/miR-126 

To investigate the delivery specificity of VCAM_ab_-decorated liposome towards inflammatory HUVECs, miR-126 was labeled with cy5 to visualize its location in the cells (Figure 5A). A minimum red fluorescence signal was observed for free miR-126 and CLs-miR-126 treated HUVECs. On the contrary, Va-CLs/miR-126 treated HUVECs exhibited dramatically enhanced red fluorescence in the cytoplasm, which demonstrated that promoted liposome cell internalization rate was achieved under the guidance of VCAM_ab_. Moreover, the red fluorescence of HUVECs exhibited a dose-dependent manner, and an intensified fluorescence signal was obtained with a higher miR-126 dosage. These results proved that it is difficult for the free miR-126 to enter cells due to its electronegativity and could only enter cells with the help of liposomes. Therefore, Va-CLs can deliver miR-126 more efficiently, and VCAM_ab_-modified CLs were key to ensuring miR-126 delivery into the cells. Aiming to quantify liposome uptake by HUVECs, a flow cytometry assay was conducted (Figure 5B). Neglect for cy3 fluorescence was observed for HUVECs without any treatment, while dramatically enhanced cy3 fluorescence was observed for HUVECs with liposome treatment for 3 h, which was caused by the positive surface zeta potential of the liposome. Furthermore, higher fluorescence intensity was observed for VCAM_ab_-decorated liposomes in comparison with its non-antibody-modified counterpart, which agreed with CLSM analyses and proved that VCAM_ab_ promoted cell internalization. Thus, it is likely that the same percentage of cells underwent uptake by liposomes and immunoliposomes, but the intensity of uptake was higher for immunoliposomes due to the presence of VCAM_ab_ on the particle surface, which can be internalized in more intensity after binding to VCAM-1 receptors. The receptor inhibition targeting assay showed (Figure 5C) that for cells pre-saturated with VCAM_ab_, the average intracellular fluorescence intensity of Va-CLs/miR-126 decreased significantly by 39.9%, while the average intracellular fluorescence intensity of non-VCAM_ab_-modified CLs/miR-126 transfected cells did not change significantly before and after VCAM_ab_ saturation.

### 3.6. Cytotoxicity

The cytotoxicity of Va-CLs/miR-126, Va-CLs, CLs/miR-126, CLs and free miR-126 was measured by MTT assay (Figure 6). No significant cytotoxicity was observed when the concentration reached 10^2^ nM, proving their relative safety in vitro of them. Va-CLs and CLs had no obvious cytotoxicity at the concentration range of 10^−10^ to 1 nM. The cell viability of Va-CLs and CLs decreased at the concentration of 10^2^ nM, and there was certain toxicity with a significant difference (* *p* < 0.05). 

### 3.7. Expression of VCAM-1 on HUVECs 

To further confirm the role of Va-CLs/miR-126 in regulating inflammation-related factors during endothelial dysfunction, the expression level of VCAM-1 was measured. The expression of VCAM-1 protein was observed by immunofluorescence staining (Figure 7A). Consistent with the results of the Western blot assay, LPS upregulated the expression level of VCAM-1 protein, and after protection by Va-CLs/miR-126, CLs/miR-126 and miR-126, all significantly downregulated the expression level of VCAM-1 compared with the model group, in which the effect of Va-CLs/miR-126 was more obvious. The experimental results showed that after protection by Va-CLs/miR-126, CLs/miR-126 and miR-126, the expression level of VCAM-1 was reduced to different degrees compared with the model group, among which the effect of Va-CLs/miR-126 was more obvious, while the effect of free miR-126 in reducing VCAM-1 was not significantly different compared with the model group. The above results suggest that Va-CLs can more effectively deliver miR-126 into the cell to downregulate VCAM-1 protein (Figure 7B). The levels of VCAM-1 mRNA were also analyzed. The experimental results showed that after protection by Va-CLs/miR-126, CLs/miR-126 and miR-126, the VCAM-1 mRNA expression level was reduced to different degrees compared with the model group, among which the effect of Va-CLs/miR-126 was more obvious, which was consistent with the WB results (Figure 7C).

### 3.8. Determination of Cytokine Secretion by ELISA 

The levels of inflammatory factors including TNF-α, IL-1β and IL-6 were detected by ELISA to evaluate the secretion of inflammatory factors by HUVECs under different treatments. Compared with the control group, TNF-α, IL-1β and IL-6 levels of HUVECs with LPS treatment were significantly increased. This suggested that LPS could induce HUVECs to secrete inflammatory cytokines. The elevated inflammatory cytokines expression levels were significantly suppressed under the treatment of miR-126, CLs/miR-126 and Va-CLs/miR-126 with equivalent miR-126 concentration. For CLs/miR-126 treated HUVECs, an enhanced anti-inflammation effect was exhibited relative to free miR-126 treated HUVECs, as significantly lowered inflammatory cytokines were observed after the treatment. This proves that the liposome structure can effectively protect RNA from enzymatic degradation in physiological conditions. Furthermore, Va-CLs/miR-126 treated HUVECs displayed significantly lowered TNF-α, IL-1β and IL-6 levels relative to CLs/miR-126 treated HUVECs (Figure 8), which demonstrated promoted cell uptake rate and gene transfection efficiency under the guidance of VCAM_ab_. 

### 3.9. Analysis of Inflammation in Mice

The results of HE staining of thoracic aortic tissue in mice are shown in Figure 9A. In the control group, the cells of the thoracic aorta were closely arranged without obvious cell necrosis, etc. In the LPS group, fibrinoid degeneration and necrosis of the outer membrane, fracture of the elastic lamina, neutrophil and lymphocyte infiltration in the perivascular intima, mesentery and outer membrane, giant cell reaction and cystic dilatation were seen. The Va-CL/miR-126 and CLs/miR-126 groups significantly improved these pathological changes and restored the cell morphology, but the CLs/miR-126 group still showed neutrophil and lymphocyte infiltration in the perivascular intima, mesentery and epithelium. An image analysis of IHC revealed that the positive area of VCAM-1 in the sections of the LPS group was more remarkable compared to that of the other groups, and the positive area of Va-CLs/miR-126 group was decreased more significantly compared to CLs/miR-126 group (Figure 9A). The levels of IL-1β and IL-6 in mice serum were measured using ELISA (Figure 9B). LPS significantly upregulated the levels of IL-1β and IL-6 in mice serum. After treatment with Va-CLs/miR-126 and CLs/miR-126, the levels of IL-1β and IL-6 in serum were reduced to different degrees, with Va-CLs/miR-126 more significantly reducing the levels of IL-1β and IL-6 (*p* < 0.01). 

### 3.10. In Vivo Imaging Experiment

To study the inflammatory targeting ability in vivo, the cy5-labeled miR-126 nanoplatform system was injected into mice through the tail vein. The ex vivo images of the main organs were taken 3 h after injection (Figure 10A,B). The fluorescence signal in the thoracic aorta was significantly elevated in the Va-CLs/miR-126 group relative to the CLs/miR-126 group (Figure 10C), while the fluorescence signal in other organs was not significantly different between Va-CLs/miR-126 and CLs/miR-126, which may be due to the high expression of VCAM-1 at the site of vascular injury. This result demonstrates the excellent targeting and accumulation ability of the miR-126 nanoplatform at sites of vascular inflammation. In addition, the cy5-labeled miR-126 nano-delivery platform was injected into inflamed zebrafish through the tail vein to examine the targeting of the delivery platform to sites of inflammatory damage in the zebrafish vascular endothelium (Figure 10D). Compared with the CLs/miR-126 group, the Va-CLs/miR-126 group showed a stronger fluorescent signal in the thoracic aorta at 6 h post-injection, which may be due to the high expression of VCAM-1 at the site of injury, while from 12 h onwards, the results of the two groups were not significantly different, demonstrating good targeting and accumulation ability of this nanoplatform at the site of inflammation.

## 4. Discussion

Despite the tremendous advances in new technologies, such as interventional therapy with advances in medicine, morbidity and mortality, cardiovascular disease is still prevalent. Meanwhile, the prevention and treatment of vascular endothelial dysfunction have been recognized as important measures to prevent and treat cardiovascular diseases [4,5,6]. Inflammation in vascular endothelial dysfunction is a participant and regulator of acute or chronic CVD [46,47] characterized by the secretion of various pro-inflammatory factors and the expression of specific cell adhesion molecules on endothelial cells [48,49,50,51]. Therefore, we hypothesize that these molecules can serve as specific targets that attract specific target heads to bring the delivery system to the site of developing inflammation, making endothelial cell inflammation-specific delivery a promising approach for drug therapy in cardiovascular disease. Since VCAM-1 is mainly involved in the development of vascular inflammation, it has the advantage of being a specific target of vascular inflammation in cardiovascular diseases [52], thus offering the possibility of designing endothelial inflammation targeted delivery systems.

Since vascular cell adhesion molecules are involved in the development of cardiovascular disease, miR-126, which can affect its protein precursor mRNA, is expected to be used as a model drug for the design of inflammatory endothelial cell-targeted delivery systems [53,54,55]. In order to achieve high therapeutic efficiency of RNA, accurate delivery and accumulation of RNA in targeted tissues/organs are of great significance for delivery systems [19]. Antibodies have been at the forefront of the development of antibody-based therapies as targets with the advantages of being highly targeted, less toxic, and free of residuals and other side effects, and contributing to the therapeutic trend of selective targeting potential that is increasingly being exploited by delivery systems active targets [56]. VCAM_ab_ can complement VCAM-1 on the surface of inflamed vascular endothelial cells in a molecular pattern, which can enhance the targeting of delivery. Therefore, in this study, VCAM_ab_-modified cationic liposomes were designed to enhance the targeted delivery of miR-126, thus reducing the enrichment of monocytes and leukocytes in vascular endothelial cells and exerting certain synergistic therapeutic effects [57]. In this study, the high expression of VCAM-1 is key to achieving targeted delivery of miR-126 in Va-CLs/miR-126; therefore, we first validated the inflammatory model of high VCAM-1 expression in HUVECs and drew on a well-established laboratory model of inflammation in the thoracic aorta of mice.

It was reported that conventional liposome-mediated gene delivery systems have difficulties in terms of RNA loading capacity, etc. [58,59,60,61]. However, our experimental results showed that DOTAP/cholesterol liposomes can overcome these problems by using the positive charge carried by the cationic liposomes themselves to electrostatically bind to the negatively charged phosphate groups in the nucleic acid structure, thus wrapping the RNA in a stable and efficient manner, which is consistent with those reported in the literature [62]. In this study, the effective coupling of antibodies to cationic liposomes was the key to preparation. We chose DOTAP and Chol with a molar ratio of 3:1 and a 5% molar ratio of PEG-functionalized phospholipids to prepare cationic liposomes. In addition, we introduced the functional phospholipid DSPE-PEG_2000_-NHS during the preparation of cationic liposomes to provide the reactive ester group required for the coupling reaction, which reacted with the primary amino group provided by the antibody to form a stable amide bond [63,64], so that antibodies were coupled to the surface of the cationic liposomes to obtain Va-CLs drug delivery system. The coupling method used in this study is simple and mature, and the by-products can be easily removed. In addition, the antibody is attached to the end of the PEG chain of the cationic liposomes, so that the RNA carrier system has both the target-seeking function of the antibody and the protective function of the PEG [19,65,66]. The results of SDS-PAGE experiments showed that after reduction treatment, the antibodies were reduced to obtain long and short chains [42], and the molecular weights of both long and short chains were slightly increased compared with the free antibody, thus further proving that the antibody was bonded to cationic liposomes, and the molecular weights of both long and short chains were increased because the liposomes contained PEG long chains. Agarose gel experiments showed that the immunoliposome can effectively load miR-126. There are two forms of miR-126 loading in the delivery system: one part of miR-126 is loaded into the hydrophilic core of the lipid delivery system and the other part of miR-126 is bound to the surface of the lipid delivery system by electrostatic adsorption to neutralize the positive charge on the surface of the delivery system [67]. When the molar ratio of DOTAP/miR-126 for Va-CLs/miR-126 and CLs/miR-126 (supporting information given in Appendix A) was greater than 32:1, the gray value of the bands tended to the equilibrium value and there were no obvious bands at the corresponding positions of miR-126, which indicated that miR-126 was completely loaded into Va-CLs.

It is well-known that effective cellular internalization is a prerequisite for RNA to exert its gene therapy effects. Since repulsion between negatively charged RNA and cell membrane hinders effective cellular uptake of RNA, different strategies are used to enhance cellular internalization [67,68]. Cationic liposomes are widely used as non-viral gene carriers for gene therapy and are considered to be safe, biocompatible, easy to prepare and provide good delivery efficiency [69]. It has been shown that cationic liposomes, upon arrival in vivo, can interact with endosomal membranes rich in anionic lipids through electrostatic interactions, followed by fusion lipids (DOTAP) to convert the lipid phase into an inverted hexagonal conformation. The membrane fusion process is achieved by inserting the fused lipid into the endosomal membrane [70]. In the in vitro experiments, we used LPS to establish the inflammation model. After the time and concentration of LPS-treated HUVECs were studied in the literature and pre-laboratory experiments [43,44,45], 8 h was chosen as the modeling time, the LPS concentration was 1 μg/mL, and the expression of VCAM-1 was approximately two times higher than that of normal controls. In in vitro gene silencing, we assessed the miR-126 gene silencing effect by laser confocal microscopy and Western blotting to verify the aggregation sites and protein levels upon entry into the cell and analyzed the suppressed expression of VCAM-1. Laser confocal results showed that miR-126 aggregates in the cytoplasm, which is consistent with the location of miRNA action. Antibody-modified cationic liposomes promote cellular uptake of miR-126, allowing more miR-126 to enter the cytoplasm, suggesting that anti-VCAM-1 targets inflammatory cells and promotes their entry into cells. The flow cytometry results were consistent with the laser confocal results. CLs have good biocompatibility and can be adsorbed by negatively charged cell membranes, and then deliver RNA into cells through membrane fusion or endocytosis. Antibodies in Va-CLs are specific and targeted and can selectively promote the delivery of nucleic acid drugs to the target site and enrich them at the site of action, thus solving the problems of large molecular weight of RNA and difficulties in cell entry, which is a good delivery system for RNA. 

WB was reflected in the efficacy, and the miR-126-loaded immunoliposomes had more effect on downregulating VCAM-1 than the miR-126-loaded liposome group. The antibody-modified cationic liposomes exerted an inflammatory targeting effect, which can enrich and promote the internalization of miR-126 at inflammatory cells, and exerted a stronger gene silencing effect. Therefore, the downregulation effect on VCAM-1 was more obvious. In addition, it is known from ELISA that miR-126 can not only downregulate the expression of VCAM-1 but it can also improve inflammation and reduce the secretion of related inflammatory factors by downregulating the expression of VCAM-1. The results also confirmed that the immunoliposomes loaded with miR-126 had a more pronounced downregulation effect on IL-1β, IL-6, TNF-α and other related inflammatory factors, indicating that the antibody exerts a targeting effect to enrich miR-126 at inflammatory cells to deliver more miR-126, resulting in a reduction of inflammatory factor expression to improve inflammation.

In vivo experiments are less convincing than in vitro experiments because of the complexity of the in vivo environment, so we also performed in vivo experiments to ensure the targeting effect of the VCAM-1 monoclonal antibody. In the in vivo targeting experiments, miR-126-loaded immunoliposomes and miR-126-loaded liposomes were injected intravenously under the premise of LPS-induced inflammation in the thoracic aorta of mice and inflammation in zebrafish, and the former were more enriched in miR-126 at the site of vascular inflammation due to the presence of the antibody target, which exerted the targeting effect of VCAM-1 antibody at the site of inflammation. The above results further confirmed that VCAM-1 antibody-coupled cationic liposomes loaded with miR-126 increased the enrichment of miR-126 at sites of vascular inflammation and improved the transfection efficiency of miR-126 in vitro and in vivo, resulting in better inflammatory therapeutic effects. As a therapeutic delivery method for VCAM-1, high expression in cardiovascular disease shows considerable promise for application.

## 5. Conclusions

In conclusion, a cationic liposome formulation targeting inflammation-injured vascular endothelial cells with antibodies was developed. To the best of our knowledge, the present study reports immunoliposome complexes of antibody-RNA conjugates targeting inflammation-injured thoracic aorta and vascular endothelial cells as a therapeutic approach. Our results confirm that LPS modeling can provide a good reference for targeting vascular inflammatory injury as a delivery model, marking the successful preparation of a disease model for targeting delivery systems at sites of vascular inflammatory injury. VCAM-1 antibody-modified cationic liposomes aggregated more efficiently at the site of inflammation than non-targeted cationic liposomes. In vitro results showed that the antibody-bound targeted liposomes further mediated cellular uptake and intracellular drug delivery. Furthermore, Va-CLs-based delivery platforms appear to be potential novel nanocarriers for RNA delivery to VCAM-1 overexpression sites, providing new options for miRNA delivery and cardiovascular disease treatment.

## Figures and Tables

**Figure 1 pharmaceutics-15-01379-f001:**
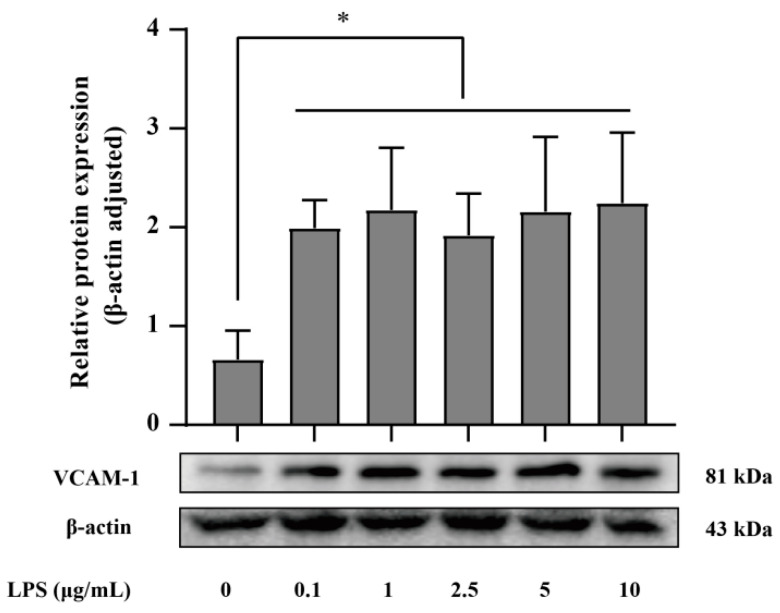
Expression of VCAM-1 in HUVECs cells treated with LPS for 8 h. (* *p* < 0.05).

**Figure 2 pharmaceutics-15-01379-f002:**
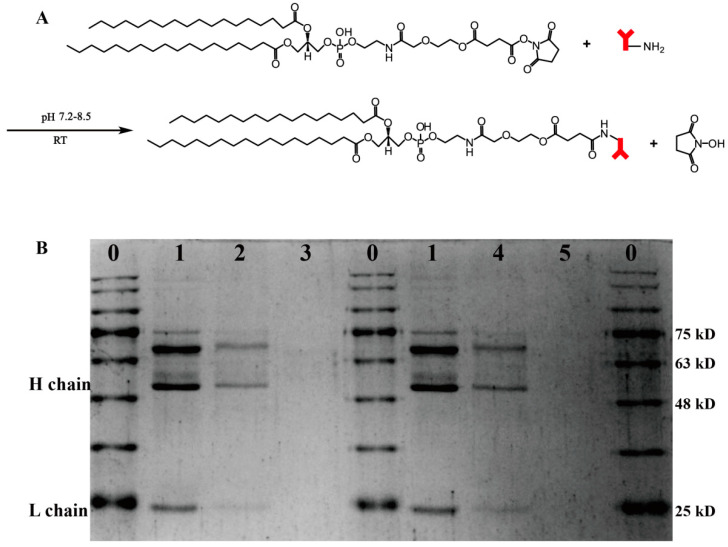
Preparation of liposomes coupled with VCAM_ab_. (**A**) Schematic illustration of the coupling of anti-VCAM-1 to liposomes. (**B**) SDS-PAGE electrophoresis under reducing conditions: 0, protein marker; 1, anti-VCAM-1; 2, Va-CLs/miR-126; 3, CLs/miR-126; 4, Va-CLs; 5, CLs. (Note: anti-VCAM-1 concentration in 1 is not comparable to the antibody levels in 2 and 4).

**Figure 3 pharmaceutics-15-01379-f003:**
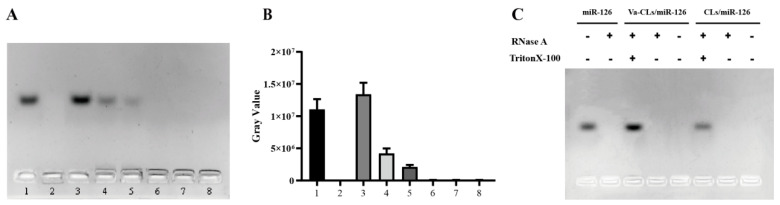
The results of agarose gel electrophoresis. (**A**) Identification of miR-126 conjugates using gel retardation assay. 1, miR-126; 2, Va-CLs; 3–8, DOTAP:miR-126 = 16:1, 24:1, 32:1, 40:1, 48:1 and 56:1. (**B**) The gray value of each miR-126 band was calculated using Image J. (**C**) Nucleic acid degradation experiment of Va-CLs/miR-126 complexes.

**Figure 4 pharmaceutics-15-01379-f004:**
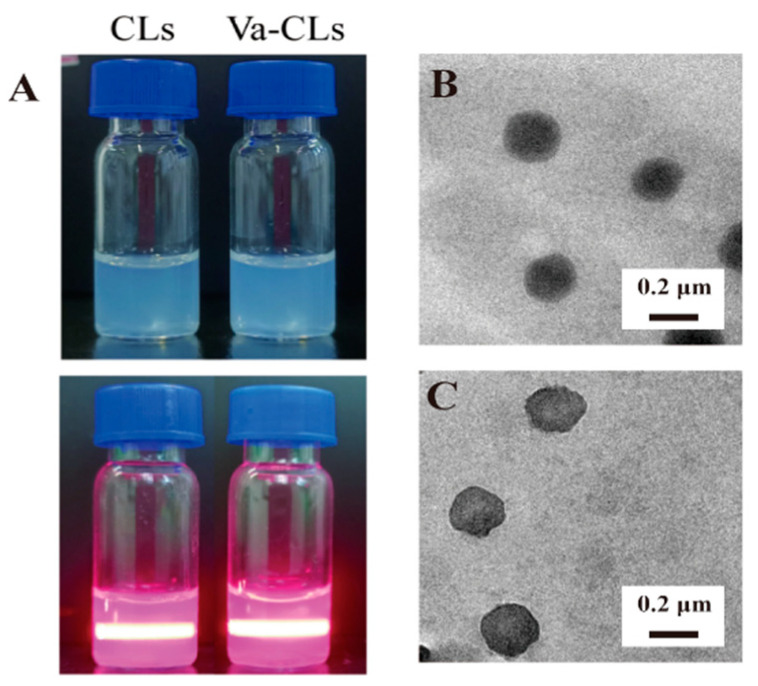
(**A**) The Tyndall effect of different solutions obtained via the complexation of CLs and Va-CLs. (**B**) TEM images of CLs/miR-126. (**C**) TEM images of Va-CLs/miR-126.

**Figure 5 pharmaceutics-15-01379-f005:**
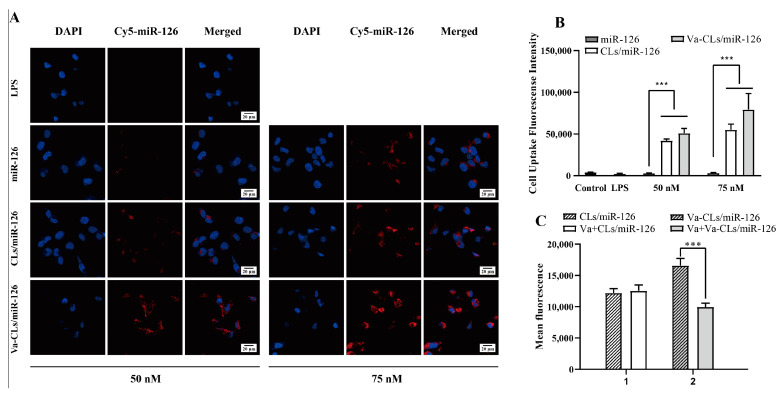
(**A**) Confocal microscope images of HUVECs after 3 h treatment with unmodified liposome or Va-CLs. The red color represents cy5-miR-126 and the blue color represents the DAPI staining of the nucleus. (**B**) Flow cytometry analysis of LPS-induced HUVECs after 3 h treatment with cy3-labeled miR-126-loaded Va-CLs and unmodified CLs (*** *p* < 0.001). (**C**) Receptor inhibition assay that compares mean fluorescence intensity of nanoparticles using flow cytometry analysis (*** *p* < 0.001). Each bar represents mean fluorescence intensity ± S.D. (n = 3).

**Figure 6 pharmaceutics-15-01379-f006:**
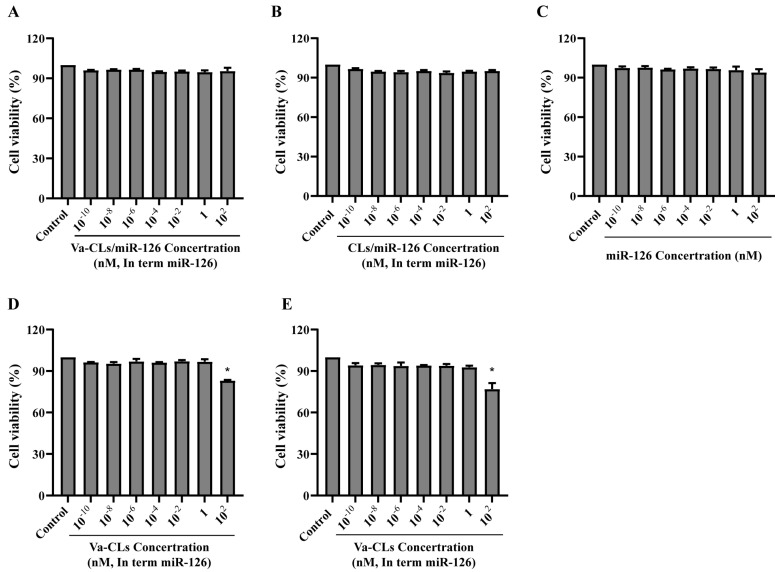
MTT assay for cytotoxicity on HUVECs cell viability. (**A**) Va-CLs/miR-126 cytotoxicity. (**B**) Cytotoxicity of CLs/miR-126. (**C**) Cytotoxicity of miR-126. (**D**) Va-CLs cytotoxicity. (**E**) Cytotoxicity of CLs. * *p* < 0.05 vs. Control.

**Figure 7 pharmaceutics-15-01379-f007:**
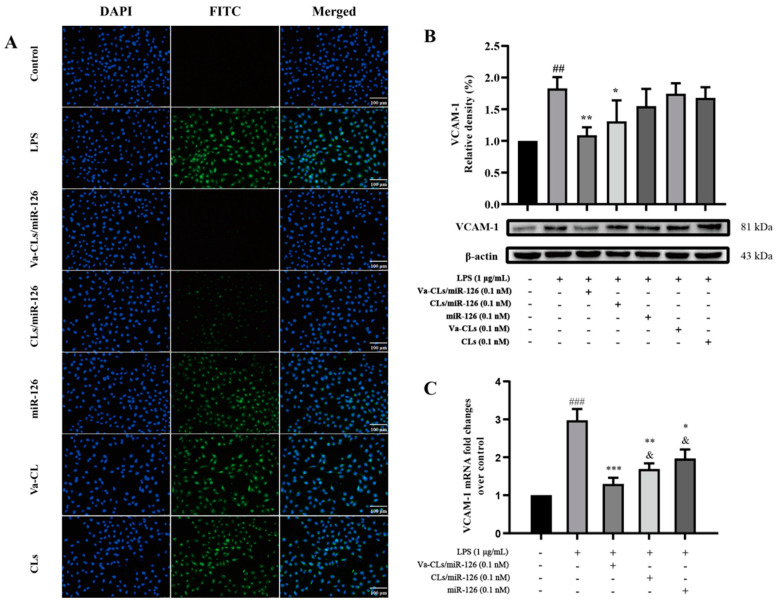
(**A**) Immunofluorescence was used to observe the expression of VCAM-1 (10×). (**B**) Va-CLs/miR-126 attenuated the expression of VCAM-1 of LPS-induced HUVECs injury (^##^ *p* < 0.01 vs. Control, ** *p* < 0.01 and * *p* < 0.05 vs. LPS.). (**C**) Va-CLs/miR-126 attenuated the expression of VCAM-1 mRNA of LPS-induced HUVECs injury (^###^ *p* < 0.001 vs. Control, *** *p* < 0.001, ** *p* < 0.01, * *p* < 0.05 vs. LPS, and ^&^ *p* < 0.05 vs. Va-CLs/miR-126.).

**Figure 8 pharmaceutics-15-01379-f008:**
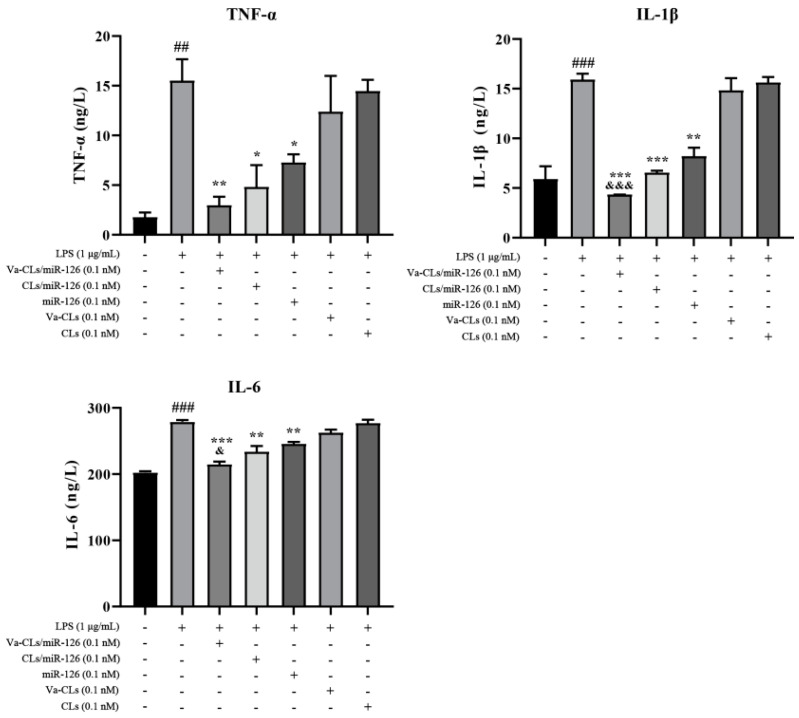
TNF-α, IL-1β and IL-6 expression level of inflammatory HUVECs after being treated with Va-CLs/miR-126 and CLs/miR-126 with equivalent miR-126 concentration (10^−1^ nM). (^###^ *p* < 0.001, ^##^ *p* < 0.01 vs. Control, *** *p* < 0.001, ** *p* < 0.01, * *p* < 0.05 vs. LPS, ^&&&^ *p* < 0.001, and ^&^ *p* < 0.05 vs. miR-126).

**Figure 9 pharmaceutics-15-01379-f009:**
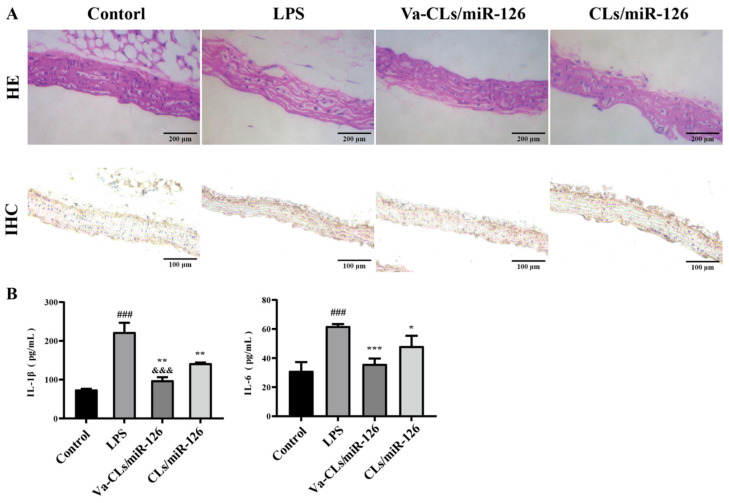
(**A**) H&E-stained histological sections (400×) and IHC-stained histological sections of mice in different groups. (**B**) Effect of Va-CLs/miR-126 and CLs/miR-126 on serum levels of IL-1β and IL-6 in LPS-induced mice (^###^ *p* < 0.001, vs. Control, *** *p* < 0.001, ** *p* < 0.01, * *p* < 0.05, vs. LPS, ^&&&^ *p* < 0.001, vs. CLs/miR-126.).

**Figure 10 pharmaceutics-15-01379-f010:**
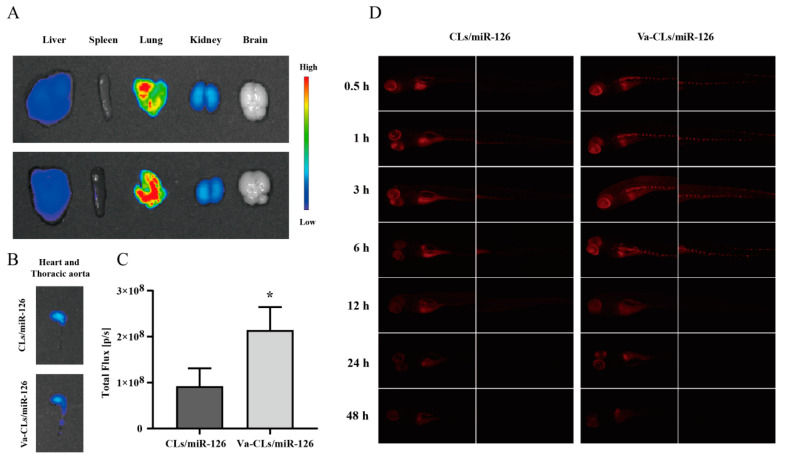
Inflammatory injury site targeting abilities of Va-CLs in vivo. (**A**) Fluorescence images of all viscera dissected from LPS-induced Kunming mice 3 h after intravenous administration with Va-CLs/cy5-miR-126 and CLs/cy5-miR-126. (**B**) Fluorescent images of heart and thoracic aortas. (**C**) Quantitative analysis of fluorescence intensity in heart and thoracic aortas. (**D**) Inflammatory injury site targeting abilities of Va-CLs/miR-126 in zebrafish. Data showed as mean ± S.D. (n = 3). ** p* < 0.05.

**Table 1 pharmaceutics-15-01379-t001:** Physicochemical characteristics of immunoliposomes.

Formulation	Particle Size/nm	Zeta Potential/mV	PDI
CLs	125.69 ± 1.92	16.13 ± 3.25	0.134 ± 0.036
CLs/miR-126	158.17 ± 0.35	11.10 ± 0.19	0.174 ± 0.028
Va-CLs	135.70 ± 1.49	10.12 ± 0.11	0.152 ± 0.012
Va-CLs/miR-126	162.19 ± 0.65	7.83 ± 0.84	0.146 ± 0.015

## Data Availability

The data presented in this study are available on request from the corresponding authors.

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
