# Peer review of "MiR-126-Loaded Immunoliposomes against Vascular Endothelial Inflammation In Vitro and Vivo Evaluation"

_pharmaceutics, 2023, doi:10.3390/pharmaceutics15051379_

Round 1
Reviewer 1 Report
Comments and Suggestions for Authors
The manuscript submitted by Tang et al. reports on the use of VCAM-1ab targeted cationic liposomes for miR126 delivery to be used in cardiovascular disease treatment. The manuscript particularly focused on cellular and in vivo experiments showing targeted liposomes ability to accumulate in vascular inflammatory sites allowing an effective miR126 delivery to vascular inflammatory endothelium. Thus, the manuscript is very interesting, easy to understand and informative. The minor correction is needed:
· Language correction and spacing mistakes (for example see lines 112, 127,131, 133,139, 398)
· Table.1 Physicochemical characteristics of immunoliposomes.
It is suggested to complete the characterization of CLs/miR-126 and Va-CLs/miR-126 (values of zeta potential).
· Please check the references (lines 387, 407, 420, 451)
· Paragraphs 2.3 and 2.4. it is suggested to combine the two paragraphs since the encapsulation of miR126 concerns both CLs/miR-126 that Va-CLs/miR-126.
Reviewer 2 Report
The manuscript of the article prepared by Yongyu Tang, Ying Chen, Qianqian Guo, Lidan Zhang, Huanhuan Liu, Sibu Wang, Yang Ding, Xingjie Wu, Xiangchun Shen and Ling Tao “MiR-126-Loaded Immunoliposomes against Vascular Endothelial Inflammation In Vitro Evaluation and In Vivo Targeting” is supposed to give the insights regarding development a miR-126 loaded immunoliposome with vascular cell adhesion molecule-1 monoclonal antibody decorated at its surface. The authors proposed that the mentioned immunoliposomes can be directly targeted to VCAM-1 at the inflammatory vascular endothelial membrane surface and achieve highly efficient treatment against inflammation response.
I am not a strong expert in the biological part of the work, and my comments mainly will address the information regarding the preparation and characterisation of liposomes. Some issues that needed to be added and addressed before publication are given below.
Major issues:
1. Lines 127-132. The authors gave very general information regarding cation liposomes without mentioning of any reference. Please, add more detailed information.
2. Please use the same terms through all the text. In line 127 authors used the term cation liposome, but in line 185 – cationic liposomes.
3. Lines 198- 199. Authors wrote” For preparing miR-126 loaded Va-CLs complexes (Va-CLs/miR-126), Va-CLs was incubated with miR-126 with Va-CLs for 30 min at 37℃”. Please correct this sentence. Also add information regarding preparation of CLs/miR-126 complex.
4. Subparagraph 2.8. Va-CLs/miR-126 and CLs/miR-126 characterization. Please add more detailed information regarding sample preparation for TEM and DLS measurements. Add missing data - Software, specifications: medium, refractive index, viscosity, dielectric constant, nanoparticles, the refractive index of materials and wavelength regarding DLS experiments.
5. Please change Figures 2B and 6 to more easily readable ones.
6. Please add information regarding VCAMab coupling efficiency to CLs containing NHS-PEG2000-DSPE. In what way is possible to detect the amount of coupled antibody? What is coupled amount?
7. Please use whole numbers without decimals in discussion about the diameters of nanoparticles.
8. Line 418. Authors wrote – “The Tyndall effect of Va-CLs and CLs were observed under light beam irradiation (Figure 4A)”. However, the legend of Fig. 4 is following “Figure 4. (A) The Tyndall effect of different solutions obtained via the complexation of CLs/miR-126 and Va-CLs/miR-126.” Please specify, do mentioned samples contain miR-126 or not? What is the content of all 4 samples demonstrated in Figure 4A?
9. Please justify the procedure for miR-126 labelling with cy5.
10. Figure 5. What means concentrations 50 nM and 75 nM. They are not mentioned in the manuscript text.
11. In the beginning of 3.5 subparagraph authors discussed about work with cy5, but in the following text about cy3. Which is the right fluorescent dye in this study?
12. Lines 612-613. Please justify the choice of lipids and their ratio used in this experiment.
Minor issues:
1. Line 133. Please remove dot between words ‘specificity. monoclonal’.
2. Lines 151,152 and 162. Please, start sentences with capitalized words.
3. Carefully check text. There is marked text – ‘Error! Reference source not found’ in lines 387, 407, 420, 451.
4. Please correct text in line 398.
5. cable 1. Use the same abbreviations for particle size.
Consequently, I do recommend accepting this manuscript for publication with major revision.
Reviewer 3 Report
Thea authors describe in detail the preparation, characterization of MiR-126-loaded immunoliposomes against vascular endothelial inflammation followed by in vitro and in vivo tests. The experiments are well explained.
Nevertheless, I have a few comments /questions to address:
11. Could you explain why did you choose to examine the 368 VCAM-1 expression levels of HUVECs by performing the incubation of HUVECs with different concentrations of LPS only for 8 h? How did you choose this time of incubation? Do you think that a longer or shorter incubation time would reveal different results?
22.Please add references: line 387 page 10; line 407 page 11; line 420 page 12; line 451 page 13
33. It is not clear the expression “the lipid complex compounded” line 401 page 11
44.Line 628: better replace “hydrophilic nucleus of the lipid delivery” with ”hydrophilic core of…”
Round 2
Reviewer 2 Report
Accept in present form